# Graphene Oxide Derivatives and Their Nanohybrid Structures for Laser Desorption/Ionization Time-of-Flight Mass Spectrometry Analysis of Small Molecules

**DOI:** 10.3390/nano11020288

**Published:** 2021-01-22

**Authors:** Seung-Woo Kim, Sunbum Kwon, Young-Kwan Kim

**Affiliations:** 1Department of Chemistry, Dongguk University-Seoul, 30 Pildong-ro, Jung-gu, Seoul 04620, Korea; swkim94@naver.com; 2Department of Chemistry, Chung-Ang University, 84 Heukseok-ro, Dongjak-gu, Seoul 06974, Korea

**Keywords:** graphene oxide, nanocomposite, surface functionalization, laser desorption/ionization, mass spectrometry

## Abstract

Matrix-assisted laser desorption/ionization (MALDI) has been considered as one of the most powerful analytical tools for mass spectrometry (MS) analysis of large molecular weight compounds such as proteins, nucleic acids, and synthetic polymers thanks to its high sensitivity, high resolution, and compatibility with high-throughput analysis. Despite these advantages, MALDI cannot be applied to MS analysis of small molecular weight compounds (<500 Da) because of the matrix interference in low mass region. Therefore, numerous efforts have been devoted to solving this issue by using metal, semiconductor, and carbon nanomaterials for MALDI time-of-flight MS (MALDI-TOF-MS) analysis instead of organic matrices. Among those nanomaterials, graphene oxide (GO) is of particular interest considering its unique and highly tunable chemical structures composed of the segregated sp^2^ carbon domains surrounded by sp^3^ carbon matrix. Chemical modification of GO can precisely tune its physicochemical properties, and it can be readily incorporated with other functional nanomaterials. In this review, the advances of GO derivatives and their nanohybrid structures as alternatives to organic matrices are summarized to demonstrate their potential and practical aspect for MALDI-TOF-MS analysis of small molecules.

## 1. Introduction

Matrix-assisted laser desorption/ionization (MALDI) technique, often combined with time-of-flight mass spectrometry (MALDI-TOF-MS), has been considered as a powerful and essential technique to analyze the intact molecular weight of high-molecular-weight compounds such as proteins [1], nucleic acids [2] and synthetic polymers [3] without their undesired fragmentation. This soft-ionization strategy has significantly contributed to the advances in chemical and biological researches based on its simple analysis process, miniscule sample consumption, high resolution, salt-tolerance, sensitivity and compatibility with high-throughput analysis [4,5]. The detailed mechanism of the MALDI process is still not fully understood, but it has been generally described by serial 3-step processes including laser energy transfer from matrix to analyte in their solid-state mixture upon laser irradiation, ionization by photochemical reaction and isolation of ionized analyte in excess matrix for mass spectrometric analysis [4]. Although MALDI-TOF-MS has been successfully applied to biological and polymer research fields for the molecular weight analysis of high-molecular-weight compounds, it cannot be directly harnessed to analyze low-molecular-weight compounds owing to the severe background interference in the low-mass region from detector saturation and/or photochemical side reactions by organic matrices such as 2,5-dihydroxybenzoic acid (DHB), α-cyano-4-hydroxycinnamic acid (CHCA), sinapic acid, and caffeic acid [6]. For addressing this issue, many efforts have been devoted to the development of an efficient alternative to organic matrices that can be used in matrix-free laser desorption/ionization time-of-flight mass spectrometry (LDI-TOF-MS) [7,8,9]. Metal [10], semiconductor [11,12,13,14], and carbon nanomaterials [15,16,17,18] have been extensively investigated as potential mediators for the LDI-TOF-MS analysis of important low-molecular-weight compounds, including amino acids, saccharides, lipids, organic pollutants, and small peptides. The laser desorption/ionization (LDI) efficiency of those nanomaterials significantly depends on their chemical composition, size and morphology [12,19]. Therefore, development of an efficient nanomaterials-based LDI-TOF-MS platform for the analysis of those small molecules has been of central interest and has attracted much research attention from the material scientists [8]. Based on those efforts, important requirements, such as high surface area, colloidal stability, laser absorption capacity, electrical and thermal conductivity and photo-thermal conversion property, have been revealed to fabricate an efficient platform for LDI-TOF-MS analysis [20,21]. In addition to those colloidal nanomaterials, various nanoporous substrates including porous silicon [22] and titania [23] have also been investigated as a chip-based analytical platform for LDI-TOF-MS analysis of small molecules. Especially, porous silicon substrates are recognized as one of the representative chip-based LDI-TOF-MS platforms because of their excellent LDI efficiency derived from the high surface area, amenable surface and uniform mass signal distribution [24,25]. The efficiency of those nanoporous substrates in LDI-TOF-MS analysis has also been enhanced by surface modification [26,27] and subsequent nanohybridization with other functional nanomaterials [28,29].

Among those various materials, carbon nanomaterials such as graphene [30,31,32,33], mesoporous carbon [34], graphene oxide (GO) [35], carbon nanotube (CNT) [36], and carbon dot (CD) [16,37] have been considered as excellent candidates because they meet most of the requirements as an efficient mediator for LDI-TOF-MS analysis of small molecules. In addition to the general requirements, the carbon materials can provide cost-effectiveness, functionalizable surface and strong affinity to various biomolecules [38] and environmental pollutants [39,40,41]. Especially, GO is a distinct carbon nanomaterial owing to its unique chemical structures composed of small segregated sp^2^ carbon domain surrounded by sp^3^ carbon matrix presenting oxygen containing functional groups [42]. Those sp^2^ carbon structures of GO derivatives play an important role in LDI-TOF-MS analysis by absorbing laser energy and converting it into thermal energy through the electron-phonon interaction for LDI of small molecules [35]. In this regard, Raman spectroscopy is a powerful and essential analytical tool to characterize the ordered and defected sp^2^ carbon structures of GO derivatives and their nanohybrid structures, which are closely related to their electron-phonon transition and then their efficiency in LDI-TOF-MS analysis [42,43,44]. GO derivatives are a complex family presenting the structural diversity depending on their synthetic and post-treatment processes. The physicochemical properties of GO derivatives greatly affect their behavior in LDI-TOF-MS analysis, and their detailed chemistry has been extensively reviewed elsewhere [45,46,47,48,49,50,51,52]. In addition, GO can be converted into graphene analogues by chemical and thermal reduction treatments for partial removal of oxygen containing functional groups, mainly hydroxyl and epoxy groups. However, there still remains residual oxygen containing functional groups on the reduced GO (RGO) because of the restricted degree of deoxygenation [52]. Thanks to these residual oxygen containing functional groups, GO and RGO derivatives can be hybridized with metal, metal oxide, and semiconductor nanomaterials by covalent and non-covalent surface modifications. The resulting nanohybrid structures can enrich thioloated, phosphorylated and/or aromatic biomolecules such as nucleic acids, amino acids, peptides and proteins through metal-thiol, metal oxide-phosphate and π-π interactions, respectively [53,54,55,56]. Based on those properties, GO and RGO derivatives and their nanohybrid structures have been actively investigated for LDI-TOF-MS analysis and exhibited a strong and versatile potential to analyze various kinds of important small molecules. There are several review articles which deal with the various nanomaterials-based matrices for LDI-TOF-MS analysis of small molecules [19]. Given the promising prospect and strong potential of GO-based nanohybrid structures for LDI-TOF-MS analysis, we think that GO derivatives and their nanohybrid structures should be solely reviewed with more detailed and comprehensive information. 

## 2. GO Derivatives for LDI-TOF-MS Analysis

Dong et al. [57] reported that graphene can be harnessed as a novel matrix for LDI-TOF-MS analysis of various small molecules, such as amino acids, polyamines, anticancer drugs, nucleosides and steroids, regardless of their polarity. It is noteworthy that the graphene used in this report was actually chemically-RGO flakes. Although the term graphene is only applicable to a single-layer of networked atomic sp^2^ carbon sheet compacted into a honeycomb lattice, it has been widely misused for RGO and few-layered graphite [58,59]. Therefore, we will use the term “RGO” rather than “graphene” throughout this review because most of the cited literature have utilized RGO for LDI-TOF-MS analysis of small molecules. In the report of Dong et Al., RGO was synthesized by using sodium dodecylbenzene sulfonate (SDBS) as a surfactant to prevent irreversible aggregation of RGO in aqueous media through the van der Waals and π-π interactions between their basal planes [60]. Despite of the surface-adsorbed SDBS on RGO, the resulting RGO exhibited many advantages for LDI-TOF-MS analysis of small molecules such as the high reproducibility, salt tolerance and applicability to the solid-phase extraction of squalene [57]. This report presents the possibility of GO derivatives as an efficient platform for LDI-TOF-MS analysis. However, it is of note that the surfactants on GO derivatives can interfere with the efficient energy transfer to analytes and solid-phase extraction, and thus the follow-up studies are generally excluded to use surfactants to prepare GO derivatives and their nanohybrid structures.

To address this issue, Zhou et al. [61] developed a RGO films-based LDI-TOF-MS platform by sequential fabrication processes including spin-coating of GO and subsequent chemical reduction by using hydrazine vapor. In this case, the RGO sheets were stably immobilized on a solid substrate and thus there was no demand of surfactants for dispersion in solvents. The resulting RGO films showed a higher efficiency and better reproducibility for LDI-TOF-MS analysis of an environmental pollutant, octachlorodibenzo-p-dioxin (OCDD), than the dispersed RGO powder. The limit of detection (LOD) of OCDD was found to be 500 pg with RGO films and the obtained signal was higher than the RGO powder. This difference was attributed to the clean surface of RGO films, and their planar and well-interconnected structures which facilitate the π-π interaction with OCDD, laser energy absorption, and energy transfer to OCDD for LDI-TOF-MS analysis. This report clearly shows that the advantages of chip-based LDI-TOF-MS platform. 

Lu et al. [62] demonstrated that the efficiency of RGO flakes in LDI-TOF-MS analysis was higher under negative ionization mode than positive ionization mode. Based on their results, the mass spectra of peptides, amino acids, fatty acids, nucleosides and nucleotides can be obtained by using RGO flakes under both positive and negative ionization modes, but there was a clear difference in mass spectra obtained under positive and negative ionization modes. The mass spectra obtained under positive ionization mode were composed of many kinds of multiple cationic adducts with proton and alkali metals such as [M + H]^+^, [M + Na]^+^, [M + K]^+^, [M + 2Na − H]^+^ and [M + Na + K − H]^+^. By stark contrast, the mass spectra obtained under a negative ionization mode were only composed of a single deprotonated form such as [M − H]^−^. Since the formation of multiple adducts makes identification of analytes complicated, the applicability of RGO flakes to a negative ionization mode facilitates its wide-spread usage for LDI-TOF-MS analysis [30,37,62]. 

The size and structure of graphene derivatives might also have significant influence on the LDI-TOF-MS analysis. Liu et al. [63] investigated the LDI efficiency of graphene derivatives such as graphene, GO and RGO for LDI-TOF-MS analysis of small molecules under negative ionization mode. According to the results, the graphene prepared by chemical vapor deposition showed no activity as a matrix for LDI-TOF-MS analysis, but GO and RGO flakes presented a high efficiency for LDI-TOF-MS analysis of flavonoids and coumarin derivatives. Interestingly, the LDI efficiency of GO flakes was much higher than that of RGO flakes even at 1 pmol of flavonoids, which is presumably attributed to the abundant carboxylic acid groups on GO flakes. The authors also explored size effect of GO flakes on LDI-TOF-MS analysis and found that millimeter-sized GO flakes provide a higher LDI efficiency than micrometer-sized GO flakes for the analysis of flavonoids [63]. 

Kim et al. [64] investigated the size influence of GO flakes on their fragmentation behavior during LDI-TOF-MS analysis of small molecules (Figure 1a). Considering that the fragmentation of GO flakes mainly occurs on the defect sites and labile structures composed of epoxide groups, the fragmentation of GO flakes can be strongly dependent on their lateral dimension [64]. The GO flake larger than 5 μm in their lateral dimension underwent severe fragmentation compared to the GO flakes smaller than 1 μm (Figure 1b). This observation was attributed to the increased density of defects and epoxide groups on the basal plane GO flakes with their lateral dimension [65]. The results implied that the smaller GO flakes lead to the less fragmentation during LDI-TOF-MS analysis, and this hypothesis was further confirmed with LDI-TOF-MS analysis by using nano-sized GO (NGO) flakes which obtained clear mass spectra of small molecules without interference from the fragmentation of GO flakes in low mass region [64]. By using NGO as a matrix, organic pollutants such as benzoyldibenzo-p-dioxin (BDPD), benzo[a]pyrene (B[a]P), and perfluorobutyric acid (PBA) were analyzed with LDI-TOF-MS and the LOD was determined as 15 fg, 150 fg, and 15 pg, respectively. Overall, these outcomes clearly show the potential of LDI-TOF-MS analysis to be directly utilized for investigating the chemical structure of GO derivatives.

The origin of the fragmentation behavior of GO derivatives in LDI-TOF-MS analysis was further investigated [66,67], as the true structure of GO sheets is debated owing to the recent discovery of highly oxidized species present on their surface [68]. The origin of the fragmentation could be traced to the direct fragmentation of a core graphene-like sheet or the detachment of the surface-adsorbed oxidative debris (OD). To determine the source of fragmentation, a graphene-like sheet and OD were separated from as-synthesized GO (aGO) through a base-washing process (Figure 2a), and the resulting graphene-like sheet (bwGO) and OD were subjected to LDI-TOF-MS analysis under identical conditions (Figure 2b). Comparison of LDI-TOF-MS spectra of bwGO to that of aGO showed that aGO exhibited mass peaks attributed to both pure and oxidized carbon clusters, while bwGO presented much stronger mass peaks solely due to the pure carbon clusters (Figure 2c). These results indicate that the fragmentation of GO sheets originates from both the core graphene-like sheet and the detachment of the surface-adsorbed OD; however, the separation process leads to the partial reduction of these GO sheet constituents.

In addition, the influence of OD on the efficiency of LDI-TOF-MS analysis was further investigated by comparing aGO and bwGO. The efficiency of the LDI-TOF-MS analysis of various analytes, was higher with bwGO than with aGO regardless of their chemical structure and molecular weight (Figure 3). The LOD of small molecules with bwGO was determined to be approximately 10 pmol which is lower than that with aGO (100 pmol). This demonstrates that the photo-thermal conversion efficiency of GO derivatives can be enhanced simply by removing the surface-adsorbed OD [67].

## 3. GO/CNT Hybrid Structures for LDI-TOF-MS Analysis

As discussed in the former section, GO derivatives have been extensively investigated for LDI-TOF-MS analysis of small molecules, but their direct application is relatively restricted in comparison with other carbon nanomaterials such as CNT and CD [36,37]. This is owed to their relatively low efficiency of photothermal conversion and low photochemical stability under the irradiation by high-power lasers during LDI-TOF-MS analysis [64]. The former feature results in low efficiency in LDI-TOF-MS analysis, and the latter feature leads to severe fragmentation that generates background interference in the low-mass region [64,67]. These problems are major obstacles to be overcome for the successful application of GO derivatives to LDI-TOF-MS analysis. Lee et al. [69] demonstrated that the hybrid films of GO and amine-functionalized multi-walled carbon nanotube (MWCNT-NH_2_) can be utilized as an efficient platform for the lipase-activity assay based on the LDI-TOF-MS analysis. The hybrid films were fabricated by the sequential assembly of GO and MWCNT-NH_2_ on amine-functionalized solid substrates through the strong electrostatic interaction between negatively-charged oxygen containing functional groups of GO and positively-charged amine groups of MWCNT-NH_2_. After the sequential electrostatic assembly, the resulting GO/MWCNT-NH_2_ hybrid films were thermally treated to induce formation of covalent linkages which resulted in increase in their stability during LDI-TOF-MS analysis [70]. Therefore, the hybrid films presented negligible interference in the low mass region because of their covalently connected structures. The GO/MWCNT-NH_2_ hybrid films also exhibited a high efficiency for LDI-TOF-MS analysis of lipids and fatty acids, implying that there is a synergistic effect from the interfaces between GO and MWCNT-NH_2_ that provides high laser energy absorption and photothermal conversion for LDI-TOF-MS analysis [69].

Kim et al. [71] investigated how to develop an optimized hybrid film composed of GO and MWCNT derivatives. Various combinations of GO and MWCNT derivatives were systematically exploited to prepare their hybrid films such as GO/MWCNT-NH_2_, RGO/MWCNT, and RGO/MWCNT-NH_2_ (Figure 4a). As control, the individual GO, RGO and MWCNT-NH_2_ films were also prepared to clearly reveal the possible synergistic effect from their hybridized structures. The detailed fabrication processes and structures of those hybrid films are presented in Figure 4a,b. Among the fabricated films, GO/MWCNT-NH_2_ hybrid films exhibited the highest performance for LDI-TOF-MS analysis of small molecules in terms of salt tolerance, homogeneous mass signal, sensitivity, accuracy and resolution (Figure 4c). The LOD was determined to be approximately 1 to 100 pmol. In addition, there was less fragmentation of GO/MWCNT-NH_2_ hybrid films than the other hybrid films during LDI-TOF-MS analysis, and this stability was attributed to the covalently-linked structures between the substrate, GO flakes, and MWCNT-NH_2_ [71]. This report suggests that the GO and MWCNT-NH_2_ hybrid film is a promising candidate as an efficient platform for LDI-TOF-MS analysis of small molecules. 

Then, they explored the structural effect of GO/MWCNT-NH_2_ hybrid films such as thickness and surface roughness [72]. It is well known that the oppositely charged GO flakes and MWCNT-NH_2_ can be alternatively assembled on a solid substrate by layer-by-layer (LBL) assembly technique [73,74,75]. Based on this principle, GO flakes and MWCNT-NH_2_ were successively assembled on a substrate with an alternating structure (Figure 5a), and thus the laser energy absorption capacity, thickness and surface roughness of the resulting GO/MWCNT-NH_2_ hybrid films were precisely controlled by the number of LBL assembly cycles (Figure 5a–c) [72]. The thickness and surface roughness of GO/MWCNT-NH_2_ hybrid films are directly related to their laser energy absorption capacity and interfacial area with small molecules, respectively [71,72]. By using the LBL assembled GO/MWCNT-NH_2_ hybrid films, they found that the LDI-TOF-MS analysis efficiency of GO/MWCNT-NH_2_ hybrid films increased with the number of LBL assembly up to 5 cycles, but it started to decrease with further LBL assembly. This interesting nonlinear behavior was attributed to the fragmentation of LBL assembled GO/MWCNT-NH_2_ hybrid films by too much local heating induced by laser irradiation. The results indicated that the LDI-TOF-MS analysis efficiency of GO/MWCNT-NH_2_ hybrid films is greatly affected by their physical structures such as thickness and surface roughness. The LOD of small molecules on five layered GO/MWCNT-NH_2_ hybrid films were determined to be 10 pmol for cellobiose, Leu-enkephalin and phenyl alanine, and 100 pmol for glucose, lysine and leucine.

## 4. GO/Metal Hybrid Structures for LDI-TOF-MS Analysis

Metallic nanoparticles (NPs) have been extensively explored for LDI-TOF-MS analysis of small molecules based on their well-defined synthesis, surface chemistry, high optical absorption and photo-thermal conversion [76,77,78,79,80,81]. Therefore, the hybridization of GO derivatives with metallic NPs has been considered a promising approach to the improve efficiency of LDI-TOF-MS analysis of small molecules. Kim et al. [82] demonstrated a simple approach to prepare the Au NPs/GO hybrid films. GO films were fabricated by immobilization of GO flakes on amine-functionalized glass substrates and then treated with polyallylamine hydrochloride (PAAH) to introduce primary amine groups on the GO films (Figure 6a). Five nm-sized Au NPs were incorporated onto the PAAH-functionalized GO (PAAH-GO) films by electrostatic interaction and further grown by the seed-mediated growth process (Figure 6a) [83,84]. The resulting Au NPs/PAAH-GO hybrid films exhibited high efficiency in LDI-TOF-MS analysis of small molecules without interference in low mass region (Figure 6b) and the LOD of small molecules were estimated to be 100 pmol. For systematic investigation of interfacing-structure effect between Au NPs and GO films, Au NPs on PAAH-treated glass, pyrene ethyleneglycol amine-functionalized GO (PEA-GO) and PAAH-GO films were parallelly compared as an analytical platform for LDI-TOF-MS analysis of small molecules. Interestingly, the analytical efficiency was highest on the Au NPs/PAAH-GO hybrid films among the tested films (Figure 6c), which implies that the hybridization of GO derivatives with metallic NPs considerably enhances their LDI-TOF-MS efficiency for analysis of small molecules. 

Likewise, Kuo et al. [85] investigated the hybrid structures of Au NPs and RGO flakes for LDI-TOF-MS analysis of small molecules. The Au NPs/RGO hybrid films were fabricated by using a spin-assisted LBL assembly technique and thus the resulting hybrid films presented an alternating structure of Au NPs and RGO flakes with a precisely controlled thickness by the number of LBL cycles. To find the correlation between the thickness of hybrid films and LDI-TOF-MS analysis efficiency, 2, 5, 10, 15, and 20 layers of Au NPs/RGO hybrid films were serially fabricated and applied to LDI-TOF-MS analysis of small molecules. With an increase of the number of layers, the optical absorption at N_2_ laser wavelength (337 nm) almost linearly increased and thus the LDI-TOF-MS analysis efficiency was also enhanced up to 10 LBL assembly cycles. However, the LDI-TOF-MS efficiency was diminished with further LBL assembly because the deep infiltration of analytes into the thick hybrid films impeded their efficient desorption/ionization. The optimized 10 layered Au NPs/RGO films exhibited high efficiency in LDI-TOF-MS analysis of amino acids and glutathione without interference in the low mass region at 150 pmol of various small molecules. This high performance was attributed to the combination of high thermal conductivity of RGO flakes and low heat capacity of Au NPs. 

The hybridization of Ag NPs with GO derivatives for LDI-TOF-MS analysis of small molecules was also thoroughly explored by Hong et al. [86]. They harnessed LBL assembly process of poly(diallyldimethylamonium chloride) (PDDAC) and AgNPs/RGO flakes to fabricate Ag NPs/RGO hybrid films with a controlled porosity and thickness (Figure 7). Based on their results, it was confirmed that the formation of Ag cluster ions from Ag NPs, one of main problems of metallic NPs as a LDI-TOF-MS platform, can be prevented with enhanced LDI efficiency by hybridization of RGO flakes. Interestingly, carbon ion clusters were also not formed from Ag NPs/RGO flakes hybrid films during LDI-TOF-MS analysis. This interesting behavior was presumably attributed to their highly porous structures and strong electrostatic interaction between PDDAC, Ag NPs and RGO flakes [85]. Given no interference from metal and carbon ion clusters, Ag NPs/RGO hybrid films exhibited a high applicability to LDI-TOF-MS analysis platform for carboxyl-containing small molecules such as amino acids, fatty acids, peptides and dicarboxyl-contained organic molecules (Figure 7). 

Au NPs/GO hybrid structures can also be harnessed for selective enrichment because the surface functionalization chemistry of Au NPs is well established for the enrichment of various specific targets [87]. Recently, Li et al. [88] reported the synthesis of a porous bead composed of GO and Au NPs presenting abundant binding sites for selective enrichment of N-linked glycopeptides. The porous GO beads were prepared by freeze-drying of a mixture droplet consisting of GO flakes, polyethyleneimine (PEI) and poly(ethylene glycol) diglycidyl ether (PEGDE), and thermally treated for their structural stabilization. Then, the GO beads were directly utilized as a support for direct synthesis of Au NPs on their surface and the resulting Au NPs/GO porous hybrid beads were further functionalized with glutathione to provide abundant binding sites for multivalent interaction with N-linked glycopeptides. The N-linked glycopeptides possess higher hydrophilicity than non-linked glycopeptides, and thus they can be strongly bound to the surface of Au NPs/GO porous hybrid beads. By using the porous hybrid beads, N-linked glycopeptides were successfully enriched and thus efficiently analyzed by using MALDI-TOF-MS with a high selectivity, reproducibility, and low LOD of 2 fmol.

## 5. GO/Metal Oxide Hybrid Structures for LDI-TOF-MS Analysis

The magnetic solid phase extraction (MSPE) of environmental pollutants is also a critical application of Fe_3_O_4_ NPs/RGO nanohybrid structures. In natural water, there are many kinds of antibiotics that are regarded as organic pollutants owing to their potential adverse effect on human health and ecosystems [89,90,91]. Tang et al. [92] reported that the MSPE of quinolones (QNs), which are one of the widely-used antibiotics causing a significant concern, by using Fe_3_O_4_ NPs/GO nanohybrid structures in various water sources. Since the QNs are generally present at low concentration in natural water, there is a strong demand on a facile way to enrich them for the efficient analysis [93,94]. The Fe_3_O_4_ NPs/GO nanohybrid structures were prepared by covalent incorporation of 3-aminopropyltriethoxysilane (APTES)-functionalized Fe_3_O_4_ NPs (Fe_3_O_4_ NPs-NH_2_) on the surface of GO flakes through N-(3-dimethylaminopropyl)-N′-ethylcarbodiimide (EDC)/N-hydroxysuccinimide (NHS) coupling [93]. The resulting Fe_3_O_4_ NPs-NH_2_/GO nanohybrid structures were harnessed for MSPE of 12 kinds of QNs such as enoxacin, norfloxacin, ciprofloxacin, pefloxacin, fleroxacin, gatifloxacin, enrofloxacin, levofloxacin, sparfloxacin, danofloxacin, difloxacin, and lomefloxacin [92]. For the optimization of MSPE and MALDI-TOF-MS processes, various experimental factors including acidity, extraction time, amount of adsorbent, elution solution and desorption time were systematically explored and then the enriched QNs on Fe_3_O_4_ NPs-NH_2_/GO nanohybrid structures were analyzed by using a surfaced-immobilized CHCA on SBA-15 treated with 3-APTES (SBA-15-NH_2_/CHCA) as a matrix. All QNs were successfully detected from real water samples from the Hai river by combination of MSPE and MALDI-TOF-MS analysis without background interference. This result confirmed the practical applicability of Fe_3_O_4_ NPs-NH_2_/GO nanohybrid structures as an adsorbent for MSPE of a trace amount of environmental pollutants. 

Although the previous reports only utilize Fe_3_O_4_ NPs/GO nanohybrid structures as a MSPE material [95], Fe_3_O_4_ NPs/GO nanohybrid structures can be directly applied to LDI-TOF-MS analysis after MSPE of specific target analytes because both GO and Fe_3_O_4_ NPs are an efficient material for LDI-TOF-MS analysis of small molecules [61,96]. Chien et al. [97] demonstrated that the Fe_3_O_4_ NPs/GO nanohybrid structures can be simultaneously used for MSPE and then LDI-TOF-MS analysis of glimepride, which is one of the representative medicines to reduce blood glucose levels in the patients of diabetes but also abused to induce narcotic effect. The Fe_3_O_4_ NPs/GO nanohybrid structures were prepared by incorporation of pre-synthesized Fe_3_O_4_ NPs on the surface of GO flakes, having the different number of layers through an emulsion and solvent evaporation process. By using the Fe_3_O_4_ NPs/GO nanohybrid structures, the LOD of glimepiride was determined to be as 284 pmol, 253 pmol, and 26 pmol for Fe_3_O_4_ NPs/GO nanohybrid structures prepared with single layered, 2–4 layered, and 4–8 layered GO flakes, respectively [95]. Those results clearly imply that the Fe_3_O_4_ NPs/GO nanohybrid structures are one of the promising materials for MSPE and LDI-TOF-MS analysis and the number of GO layers has critical influence on the efficiency of LDI-TOF-MS analysis on the Fe_3_O_4_ NPs/GO nanohybrid structures. 

Even without the MSPE process, metal oxide/GO nanohybrid structures can be considered an important material for LDI-TOF-MS analysis. In this regard, Kim et al. [98] demonstrated that the ZnO/RGO nanohybrid structures can be directly applied to wafer-level detection of organic contaminations with LDI-TOF-MS analysis (Figure 8a). The detection of organic contaminants on Si wafer has been a critical issue to improve the yield of semiconductor fabrication processes and thus there is a demand on the analytical tool, which can provide the localized information of chemical structures of the residual organic contaminants on the semiconductor devices [99]. Therefore, LDI-TOF-MS is a powerful analytical tool because it can provide the localized information of chemical structures which are irradiated by laser equipped in LDI-TOF-MS [100]. As a model organic contaminant, B[a]P, which is one of the common organic contaminants on Si wafer, was analyzed with LDI-TOF-MS by using ZnO/RGO nanohybrid structures [98]. The roles of ZnO and RGO in ZnO/RGO nanohybrid structures are to absorb UV light from laser and transfer the energy to analytes [101], and to enrich organic contaminants such as polyaromatic hydrocarbons from Si wafers through π-π interaction [102,103,104]. By optimizing the composition ZnO/RGO hybrid structures and their amount used, B[a]P was successfully analyzed with LDI-TOF-MS without interference in low mass region and its LOD was estimated to be 13 pmol, which was lower than the concentration of residual organic contaminations generated in the fabrication process of semiconductor devices (Figure 8b,c). The ZnO/RGO nanohybrid structure was also proven to be an effective material for LDI-TOF-MS analysis of other aromatic and aliphatic species on a semiconductor wafer.

As a summary of the important LDI-TOF-MS analytical platforms fabricated by using GO derivatives and their nanohybrid structures, the figure of merit (FOM) of those analytical platforms such as LOD values was described in Table 1. The summarized results indicated that the hybridization of CNT, metal and metal oxide nanomaterials on the surface of GO derivatives does not always guarantee the improvement of LOD values, but it can still endow their nanohybrid structures with a novel function and thus extend their analytical applicability according to the purposes. The strengths, weaknesses, opportunities and threats (SWOT) analysis was also carried out on the basis of the reviewed literature (Table 2). GO derivatives and their nanohybrid structures have exhibited many advantages including high laser energy absorption capacity, photothermal conversion efficiency, electrical and thermal conductivity, affinity to important biomolecules and environmental pollutants, and amenable surface for functionalization and hybridization with other functional groups and nanomaterials. However, they also possess disadvantages such as laser induced fragmentation, contamination of mass spectrometer, and heterogeneous lateral dimension and chemical structures. 

## 6. Conclusions

Over the past decades, GO derivatives have been extensively investigated to develop an efficient and multi-functional LDI-TOF-MS platform based on their high laser energy absorption capacity, photo-thermal conversion efficiency, thermal conductivity, tailorable surfaces, affinity toward aromatic compounds, salt tolerance, reproducibility, and large surface area. In addition, their excellent intrinsic properties can be considerably enhanced by their surface functionalization and subsequent nanohybridization with other functional carbon, metal and metal oxide nanomaterials. Inspired by these interesting characteristics, we have systematically reviewed the synthesis, structure and property relationship, surface functionalization, assembly, and nanohybridization of GO derivatives for their efficient and diverse applications to LDI-TOF-MS analysis of important low molecular-weight compounds. Although there are many kinds of nanomaterials which have been efficiently utilized for LDI-TOF-MS analysis, GO derivatives and their nanohybrid structures can provide distinct advantages and thus they will considerably contribute to various research fields including metabolomics, environmental pollution, imaging mass spectrometry, and drug discovery. However, the potential toxicity, dangerous and toxic synthetic process and unstandardized structures of GO derivatives should be addressed for their wide-spread applications. Taken together, we believe that the GO-based nanohybrid structures can provide distinct advantages from other nanomaterials and thus will be an important, practical and functional tool for LDI-TOF-MS analysis with their steady progress.

## Figures and Tables

**Figure 1 nanomaterials-11-00288-f001:**
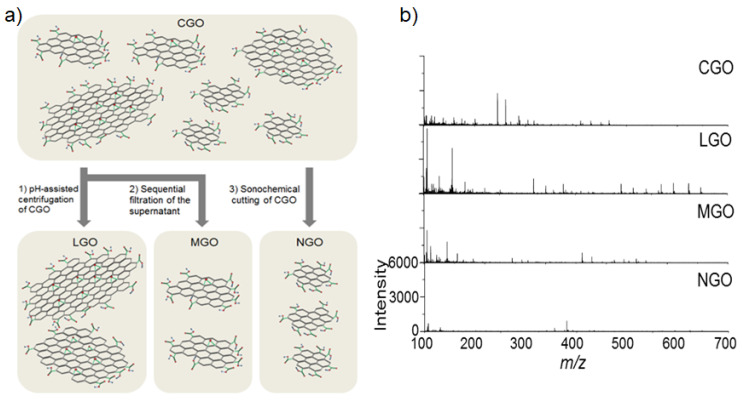
(**a**) A schematic diagram of size-fractionalization of GO flakes depending on their lateral of dimension. (**b**) LDI-TOF-MS spectra of small molecules obtained by using size-fractionalized GO flakes. Adapted with permission from ref. [64]. Copyright 2015 Wiley-VCH.

**Figure 2 nanomaterials-11-00288-f002:**
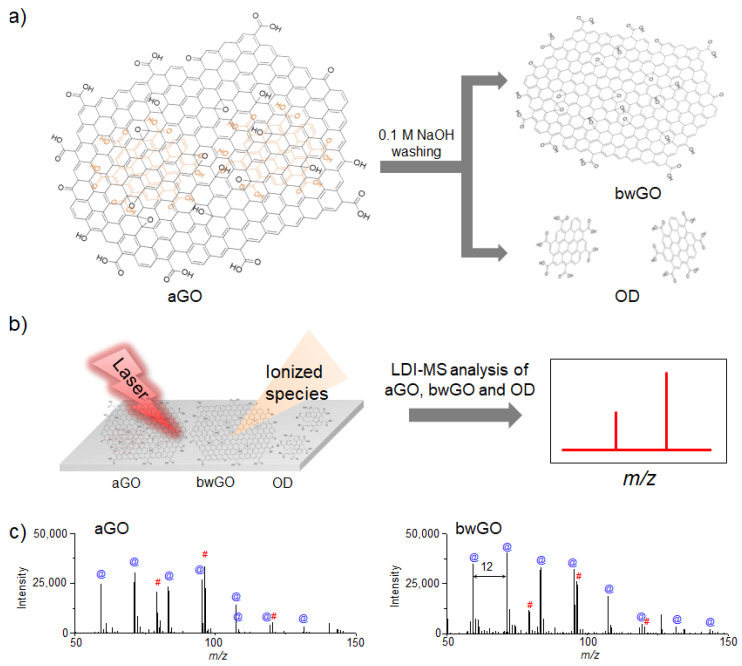
Schematic diagrams of (**a**) the separation process of OD and bwGO from aGO by washing with 0.1 M NaOH, and (**b**) LDI-TOF-MS analysis process of the obtained aGO and bwGO. (**c**) LDI-TOF-MS spectra of aGO and bwGO. The symbol @ in blue color corresponds to the carbon cluster ions and the symbol # in red color corresponds to the oxidized carbon cluster ions. Adapted from ref. [66] with permission from The Royal Society of Chemistry.

**Figure 3 nanomaterials-11-00288-f003:**
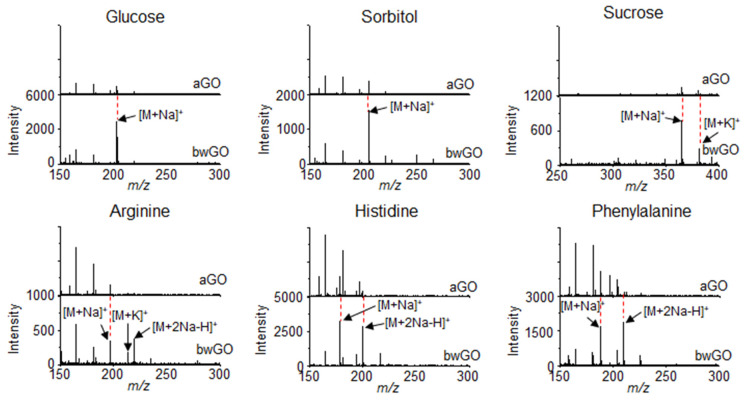
LDI-TOF-MS spectra of small molecules such as glucose, sorbitol, sucrose, arginine, histidine, and phenylalanine obtained with aGO and bwGO. Reproduced with permission from ref. [67]. Copyright (2019) Japan Society for Analytical Chemistry.

**Figure 4 nanomaterials-11-00288-f004:**
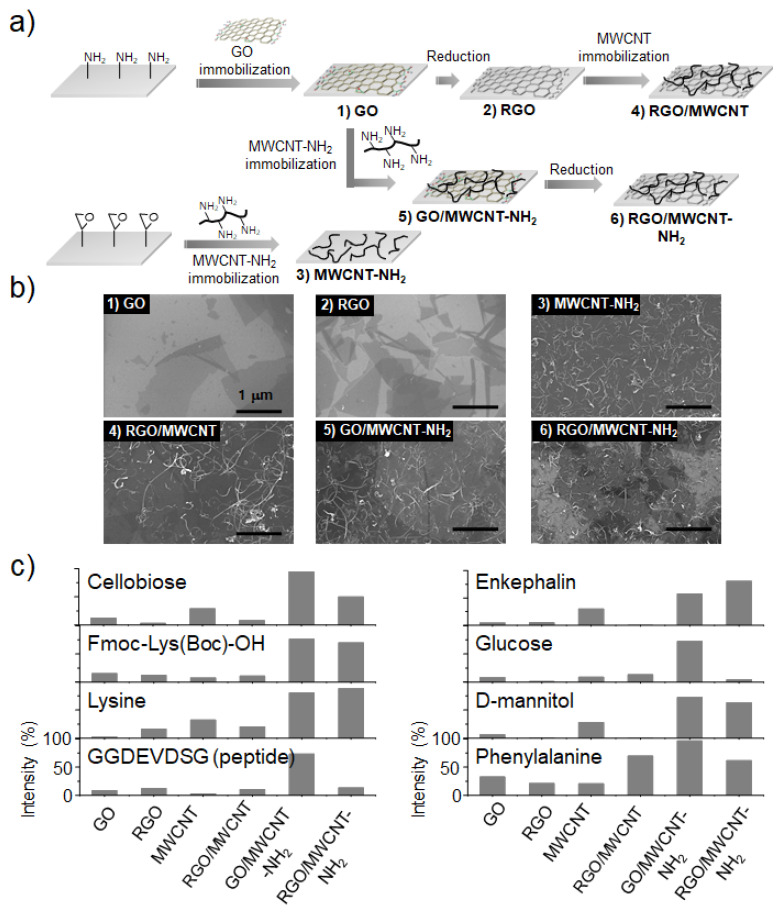
(**a**) Fabrication processes and (**b**) scanning electron microscopy (SEM) images of GO, RGO, MWCNT-NH_2_, RGO/MWCNT, GO/MWCNT-NH_2_ and RGO/MWCNT-NH_2_ nanohybrid films. (**c**) The relative mass signal intensities of various small molecules such as cellobiose, Leu-enkephalin, Fmoc-Lys(Boc)-OH, phenylalanine, glucose, lysine, D-mannitol and GGDEVDSG peptide. Adapted with permission from ref. [71]. Copyright 2011 American Chemical Society.

**Figure 5 nanomaterials-11-00288-f005:**
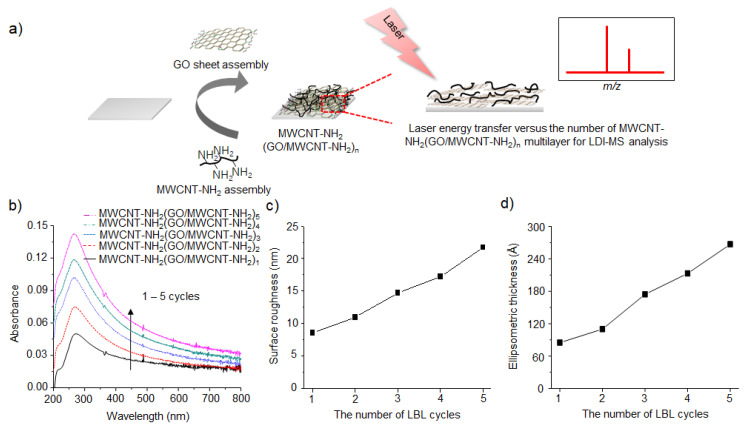
(**a**) Fabrication process, (**b**) UV-Vis absorption spectra, (**c**) surface roughness and (**d**) ellipsometric thickness of GO and MWCNT-NH_2_ nanohybrid films prepared with the different number of LBL assembly cycles. Adapted with permission from ref. [72]. Copyright 2012 American Chemical Society.

**Figure 6 nanomaterials-11-00288-f006:**
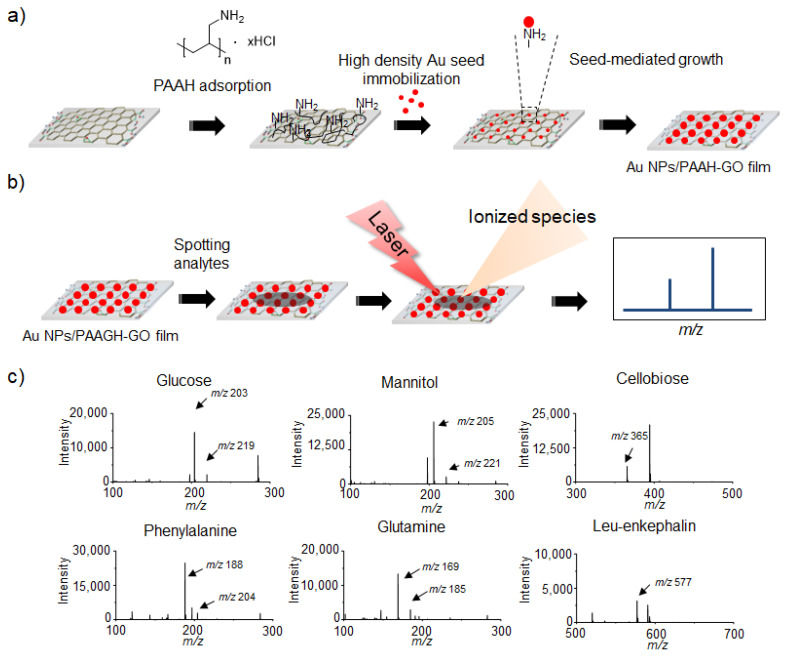
(**a**) Preparation process of Au NPs/PAAH-GO films and (**b**) LDI-TOF-MS analysis process of small molecules on Au NPs/PAAH-GO films. (**c**) LDI-TOF-MS spectra of small molecules such as glucose, mannitol, cellobiose, phenylalanine, glutamine and Leu-enkephalin obtained on Au NPs/PAAH-GO films. Adapted with permission from ref. [82]. Copyright 2012 American Chemical Society.

**Figure 7 nanomaterials-11-00288-f007:**
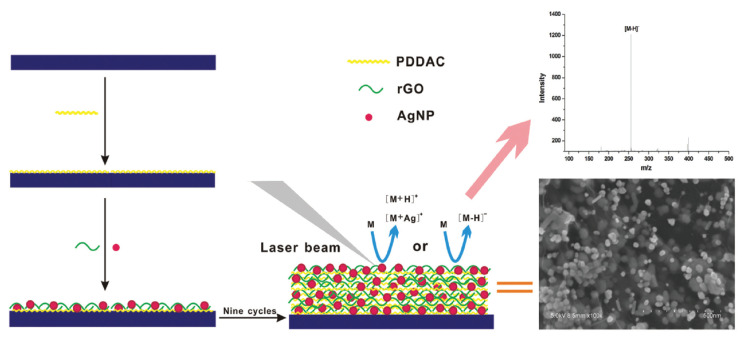
Preparation process and structure of Ag NPs/RGO nanohybrid films, and their application to LDI-TOF-MS analysis of small molecules. Reproduced from ref. [86] with permission from The Royal Society of Chemistry.

**Figure 8 nanomaterials-11-00288-f008:**
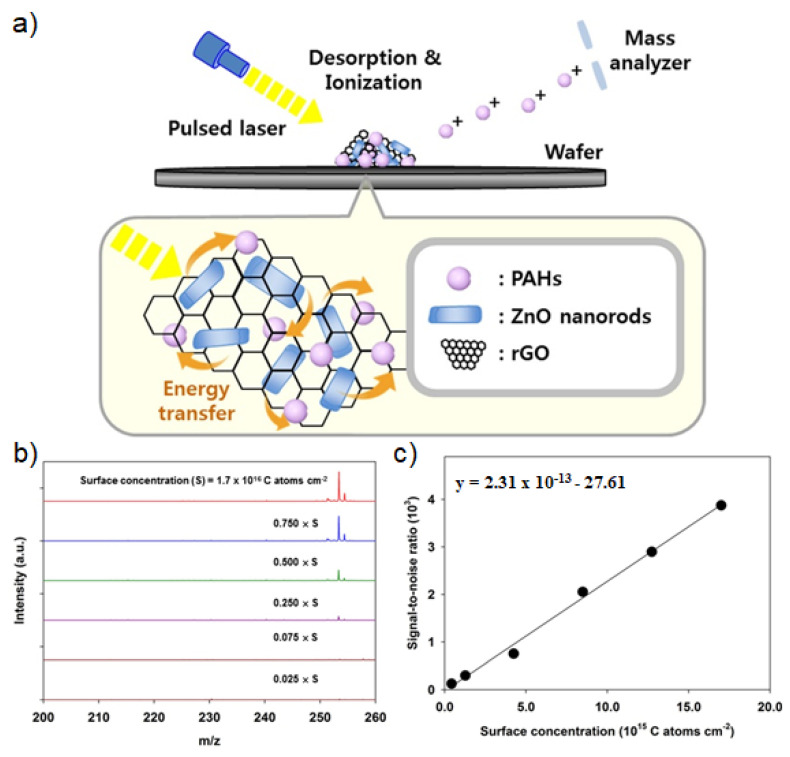
(**a**) Schematic diagram of the structure and LDI-TOF-MS application of ZnO/RGO nanohybrid structures. (**b**) LDI-TOF-MS spectra and (**c**) standard concentration curve of B[a]P obtained with ZnO/RGO nanohybrid structures with different concentration. Adapted with permission from ref. [93]. Copyright (2018) Elsevier.

**Table 1 nanomaterials-11-00288-t001:** A summary of various analytes and their LOD values for different types of GO derivatives and their nanohybrid structures for LDI-TOF-MS analysis of small molecules.

Platform	Analytes	LOD	Ref.
**RGO films**	OCDD	1 pmol	[61]
**GO**	Flavonoids	1 pmol	[63]
**NGO**	BDPD	50 pmol	[64]
	B[a]P	600 pmol	
	PBA	70 nmol	
**bwGO**	Amino acids and saccharides	10 pmol	[67]
**GO/MWCNT-NH_2_ hybrid films**	Leu-enkephalin	1 pmol	[71]
	Saccharides	10 pmol	
	Amino acids	100 pmol	
**Multi-layered GO/MWCNT-NH_2_ hybrid films**	Cellobiose	10 pmol	[72]
	Leu-enkephalin		
	Phenylalanine		
	Glucose	100 pmol	
	Lysine		
	Leucine		
**Au/PAA-GO film**	Saccharides and amino acids	100 pmol	[82]
**LBL assembled Au NPs/RGO hybrid films**	Amino acids	150 pmol	[85]
**AuNPs/GO porous hybrid bead**	N-linked glycopeptide	2 fmol	[88]
**Fe_3_O_4_@graphene oxide nanocluster**	Glimepiride	26 pmol	[97]
**ZnO-RGO hybrid**	B[a]P	13 pmol	[98]

**Table 2 nanomaterials-11-00288-t002:** A SWOT analysis result of GO derivatives and their nanohybrid structures for LDI-TOF-MS analysis of small molecules.

Strengths	Weaknesses
- Optical absorption capacity. - Photothermal conversion efficiency - Electrical and thermal conductivity - Tailorable surface for hybridization with metal, metal oxide, and semiconductor nanomaterials. - Affinity to biomolecules and environmental pollutants	- Fragmentation induced by laser irradiation - Contamination of mass spectrometer - Heterogeneous lateral dimension and chemical structures
Opportunities	Threats
- Applicability to metabolomics - Applicability to environmental monitoring - Compatibility with high-throughput analysis - Imaging mass spectrometry	- Potential toxicity of graphene derivatives - Dangerous and toxic chemicals used for synthesis of graphene derivatives - Unstandardized synthetic process and properties of graphene derivatives

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
