# Peer review of "Graphene Oxide Derivatives and Their Nanohybrid Structures for Laser Desorption/Ionization Time-of-Flight Mass Spectrometry Analysis of Small Molecules"

_nanomaterials, 2021, doi:10.3390/nano11020288_

Round 1
Reviewer 1 Report
In this manuscript, authors report the advances of GO and related materials and demonstrate their potential for MALDI-TOF-MS analysis of small molecules
Overall speaking, this manuscript has presented good revision work with potential impact in the field, however it has no considered some relevant aspects.
GO and rGO are a complex family of products, in which the production method and chemistry has strong influence (discussed previously by author [In Ref 53 (https://doi.org/10.1002/chem.201404067)); however, relevant reviews are missing: see for example: Chem. Soc. Rev., 2010, 39, 228-240, Chem. Rev. 2012, 112, 11, 6027–6053, J. Mater. Chem. C, 2020,8, 1517-1547; Nat Commun 11, 1566 (2020); Nanoscale, 2017,9, 9562-9571, https://doi.org/10.1016/j.jsamd.2017.09.003.
The surface chemistry and chemical structure of GO and rGO are key factors as described in bibliography. Moreover, author comment the presumable attributed the effects of carboxylic groups of GO vs rGO in the LDI efficiency. For these reasons in is needed to describe the surface chemistry of GO and rGO systems and correlate the preparation of the hybrid structures for MALDI-TOF-MS analysis. Moreover, surface chemistry of GO and hybrid systems has strong influence in the LBL process and it should be commented.
Another missing relevant aspect is RAMAN spectroscopy of GO and rGO and its derivatives, due to the relevance of the different types of defects in the surface chemistry as well with the different interaction of the GO & rGO systems with different molecules. It should be considered and discuss.
Minor revision:
Ref 53 describe ultralarge GO flakes (> 500 microns), however in line 161(page 4), reference is not included.
Usually, > 5 microns GO it is not considered large lateral size: see for example: 2D Materials 7(2):022001 DOI: 10.1088/2053-1583/ab1e0a; dx.doi.org/10.1021/jp3085759 | J. Phys. Chem. C 2013, 117, 10683−10690.
It is not clear for reader the differences in nomenclature between ultralarge (milimiter scale) and large lateral size GO sheets (LGO) > 5 microns. Probably more clear differences to difference both types of GO are needed.
In page 3, it is difficult to understand paragraph from lines 86 to 97. It should be rewrite to facilitate reader understand
As mentioned already, the manuscript indeed discusses recent advances of GO/rGO systems for MALDI-TOF-MS analysis, however it needs to correlate with recent research in surface chemistry of GO/rGO before to be considered for publication in Nanomaterials.
Author Response
Referee 1:
“In this manuscript, authors report the advances of GO and related materials and demonstrate their potential for MALDI-TOF-MS analysis of small molecules. Overall speaking, this manuscript has presented good revision work with potential impact in the field, however it has no considered some relevant aspects.”
“Comment 1. GO and rGO are a complex family of products, in which the production method and chemistry has strong influence, however, relevant reviews are missing: see for example: Chem. Soc. Rev., 2010, 39, 228-240, Chem. Rev. 2012, 112, 11, 6027–6053, J. Mater. Chem. C, 2020, 8, 1517-1547; Nat Commun 11, 1566 (2020); Nanoscale, 2017,9, 9562-9571, https://doi.org/10.1016/j.jsamd.2017.09.003.”
We thank to reviewer’s comment. We have added the recommended references, which describe the chemistry of GO and RGO, with a proper description “GO derivatives are a complex family presenting the structural diversity depending on their synthetic and post-treatment processes. The physicochemical properties of GO derivatives greatly affect their behavior in LDI-TOF-MS analysis, and their detailed chemistry has been extensively reviewed elsewhere [42-47]” to the revised version.
“Comment 2. The surface chemistry and chemical structure of GO and rGO are key factors as described in bibliography. Moreover, author comment the presumable attributed the effects of carboxylic groups of GO vs rGO in the LDI efficiency. For these reasons it is needed to describe the surface chemistry of GO and rGO systems and correlate the preparation of the hybrid structures for MALDI-TOF-MS analysis. Moreover, surface chemistry of GO and hybrid systems has strong influence in the LBL process and it should be commented.”
The reviewer is correct. The surface chemistry and chemical structure of GO and RGO are obviously key factors. Therefore, we have added sentences “In addition, GO can be converted into graphene analogues by chemical and thermal reduction treatments resulting in partial removal of oxygen containing functional groups, mainly hydroxyl and epoxy groups, but there are still residual oxygen containing functional groups on the reduced GO (RGO) because of the restricted degree of deoxygenation [47]. Thanks to those oxygen containing functional groups, GO and RGO derivatives can be hybridized with metal, metal oxide, and semiconductor nanomaterials by covalent and non-covalent surface modifications.” to reflect this point to the revised version.
“Comment 3. Another missing relevant aspect is RAMAN spectroscopy of GO and rGO and its derivatives, due to the relevance of the different types of defects in the surface chemistry as well with the different interaction of the GO & rGO systems with different molecules. It should be considered and discuss.”
We thank to the reviewer’s comment. Raman spectroscopy is the powerful and essential analytical tool to characterize GO derivatives and it provides information about the crystallinity of GO derivatives based on the D- and G-peaks which originate from their defected and ordered sp2 carbon structures, respectively. Although the Raman spectroscopy can provide important information about the defect structures of GO derivatives, it is not enough to clearly demonstrate their defect structures because the relative ratio of D- and G-peaks can be affected by various parameters such as lateral size, edge structure, impurities and interaction with other functional nanomaterials. As the reviewer pointed out, Raman spectra of GO-other molecule complexes and GO-based nanohybrid structures are more complicated to clearly describe because of the interfacial interaction between GO derivatives and other molecules and/or nanomaterials. Therefore, the structural changes of GO derivatives and their nanohybrid structures are systematically analyzed by comprehensive characterization results from various tools such as x-ray photoelectron, FTIR, UV-Vis and Raman spectroscopies, x-ray diffraction, thermogravimetric analysis and electron microscopy. Taken together, we think that the fragmentary discussion of Raman spectroscopy is not relevant for this manuscript owing to the diversity of the reviewed GO derivatives and their hybrid structures, and the deep discussion about the defect structures of GO derivatives is also beyond this manuscript (there are many reviews which is only focused on Raman spectroscopy of graphene derivatives; Nat. Nanotechnol. 2013, 8, 235-246; Chem. Soc. Rev. 2018, 47, 1822-1873). That is why this review only focuses on the applications of GO derivatives and their nanohybrid structures to LDI-TOF-MS analysis.
“Comment 4. Ref 53 describe ultralarge GO flakes (>500 microns), however in line 161 (page 4), reference is not included.”
We appreciate the reviewer’s comment and have added a proper reference in the revised version.
“Comment 5. Usually, > 5 microns GO it is not considered large lateral size: see for example: 2D Materials 7(2):022001 DOI: 10.1088/2053-1583/ab1e0a; dx.doi.org/10.1021/jp3085759 | J. Phys. Chem. C 2013, 117, 10683-10690. It is not clear for reader the differences in nomenclature between ultralarge (milimiter scale) and large lateral size GO sheets (LGO) > 5 microns. Probably more clear differences to difference both types of GO are needed.”
We thank the reviewer’s comment and agree with the opinion that the terms of large GO and ultralarge GO are confusing. Therefore, we have corrected the description of the lateral dimensions of GO derivatives such as ultralarge GO (millimeter scale) flakes, typical GO (~1 to 5 mm) flakes, large GO (LGO) flakes and medium-sized GO (MGO) flakes as millimeter-sized GO flakes, micrometer-sized GO flakes, the GO flakes larger than 5 mm in their lateral dimension, and the GO flakes smaller than 1 mm, respectively, in the revised version.
“Comment 6. In page 3, it is difficult to understand paragraph from lines 86 to 97. It should be rewrite to facilitate reader understand.”
We thank the reviewer’s comment. The paragraph has some confusing points and thus we have rewritten the paragraph as “Therefore, we will use the term “RGO” rather than “graphene” throughout this review because most of the cited literature have utilized RGO for LDI-TOF-MS analysis of small molecules. In the report of Dong et Al., RGO was synthesized by using sodium dodecylbenzene sulfonate (SDBS) as a surfactant to prevent irreversible aggregation of RGO in aqueous media through the van der Waals and p-p interactions between their basal planes [55]. Despite of the surface-adsorbed SDBS on RGO, the resulting RGO exhibited many advantages for LDI-TOF-MS analysis of small molecules such as the high reproducibility, salt tolerance and applicability to the solid-phase extraction of squalene [52]. This report presents the possibility of GO derivatives as an efficient platform for LDI-TOF-MS analysis. However, it is of note that the surfactants on GO derivatives can interfere the efficient energy transfer to analytes and solid-phase extraction, and thus the follow-up studies are generally excluded to use surfactants to prepare GO derivatives and their nanohybrid structures.” in the revised version.
We are grateful to the reviewers and editor for their efforts. We believe that this work can provide a fundamental insight and practical information for studying and developing an efficient LDI-MS platform by using GO derivatives and their nanohybrid structures.
With very best regards,
Young-Kwan Kim

Reviewer 2 Report
This work reviewed graphene oxide derivatives and their nanohybrid structures for laser desorption/ionization time-of-flight mass spectrometry analysis for small molecules. Compared with conventional organic matrix, nanomaterials as matrices for laser desorption/ionization time-of-flight mass spectrometry overcome the interferences in low-molecular weight region from the matrix. Graphene oxide and its derivatives, a 2D nanomaterials with large specific area, have attracted considerable attention as assisted matrix of laser desorption/ionization time-of-flight mass spectrometry for the analysis of small molecules. In my opinion, this work can be accepted for publication in Nanomaterials if the following comments can be well addressed. 1. In 2017, a work named “Nanomaterials as assisted matrix of laser desorption/ionization time-of-flight mass spectrometry for the analysis of small molecules” that also was published in Nanomaterials (Nanomaterials 2017, 7, 87), the author should be provided a simple comparison and comment for this work. 2. Many of figures were included in this work, some of them can be deleted, such as Figures 1, 2 and 8 3. Figure 3 should be presented in page 5 4. Some of the works presented in section 4 “GO/metal oxide hybrid structures for LDI-TOF-MS analysis” are not very well, the issue of this work is LDI-TOF-MS for analysis of small molecules, what are small molecules, the molecular weight is below 500 Da or 1000 Da, the molecular weights of target analytes are higher than 1000 Da (Figure 12), I think this work should be deleted.Author Response
Referee 2:
“This work reviewed graphene oxide derivatives and their nanohybrid structures for laser desorption/ionization time-of-flight mass spectrometry analysis for small molecules. Compared with conventional organic matrix, nanomaterials as matrices for laser desorption/ionization time-of-flight mass spectrometry overcome the interferences in low-molecular weight region from the matrix. Graphene oxide and its derivatives, a 2D nanomaterials with large specific area, have attracted considerable attention as assisted matrix of laser desorption/ionization time-of-flight mass spectrometry for the analysis of small molecules. In my opinion, this work can be accepted for publication in Nanomaterials if the following comments can be well addressed.”
“Comment 1. In 2017, a work named “Nanomaterials as assisted matrix of laser desorption/ionization time-of-flight mass spectrometry for the analysis of small molecules” that also was published in Nanomaterials (Nanomaterials 2017, 7, 87), the author should be provided a simple comparison and comment for this work.”
We appreciate the reviewer’s comment. Although there are several excellent review articles which deal with the nanomaterials-based matrices for LDI-TOF-MS analysis of small molecules, we think that the focused review article on GO derivatives and their nanohybrid structures is highly required considering their importance and strong potential for the LDI-TOF-MS analysis of small molecules.
To indicate this point, we have added sentences “There are several review articles which deal with the various nanomaterials-based matrices for LDI-TOF-MS analysis of small molecules [19]. Given the promising prospect and strong potential of GO-based nanohybrid structures for LDI-TOF-MS analysis, we think that GO derivatives and their nanohybrid structures should be solely reviewed with more detailed and comprehensive information.” in the revised version.
“Comment 2. Many of figures were included in this work, some of them can be deleted, such as Figures 1, 2 and 8.”
We thank the reviewer’s comment and have removed figure 1, 2 and 8 in the revised version. The figure numbers are also properly corrected.
“Comment 3. Figure 3 should be presented in page 5.”
We appreciate the reviewer’s comment and have changed the figure arrangement based on the comment of the reviewers in the revised version. The Figure 3 is designated as Figure 1, and the Figure 1 was presented in the proper page with their description.
“Comment 4. Some of the works presented in section 4 “GO/metal oxide hybrid structures for LDI-TOF-MS analysis” are not very well, the issue of this work is LDI-TOF-MS for analysis of small molecules, what are small molecules, the molecular weight is below 500 Da or 1000 Da, the molecular weights of target analytes are higher than 1000 Da (Figure 12), I think this work should be deleted.”
We appreciate the reviewer’s comment and agree with the comment. Therefore, we have removed the paragraph in the revised version.
We are grateful to the reviewers and editor for their efforts. We believe that this work can provide a fundamental insight and practical information for studying and developing an efficient LDI-MS platform by using GO derivatives and their nanohybrid structures.
With very best regards,
Young-Kwan Kim

Reviewer 3 Report
This is a very interesting review paper regarding the applications of graphene oxide derivatives and their nanohybrid structures in matrix-assisted laser desorption/ionization-time of flight mass spectroscopy.
The authors present very clear the reviewed results, with almost no spelling mistakes. However and in order for this review paper to have an impact, especially on readers that are involved in graphene research but are not so exposed to its mass spectroscopy applications more information is needed.
I propose a major revision of this manuscript in order to be published.
I suggest to the authors the following adds:
- Please explain in more details the physics of the matrix-assisted laser desorption/ionization as an analytical tool for mass spectroscopy
- Please describe the figures of merit regarding the performance of these matrices/platforms within the framework of mass spectroscopy
- Please include a summary table with all the results and the figures of merits of the reviewed papers
- I propose a SWOT analysis of the graphene-based matrices in mass spectroscopy as a potential technology
- The authors should include in the Conclusion paragraph their opinion for the perspectives and challenges of the technology they reviewed
Author Response
Referee 3:
“This is a very interesting review paper regarding the applications of graphene oxide derivatives and their nanohybrid structures in matrix-assisted laser desorption/ionization-time of flight mass spectroscopy. The authors present very clear the reviewed results, with almost no spelling mistakes. However and in order for this review paper to have an impact, especially on readers that are involved in graphene research but are not so exposed to its mass spectroscopy applications more information is needed.”
“Comment 1. Please explain in more details the physics of the matrix-assisted laser desorption/ionization as an analytical tool for mass spectroscopy.”
The reviewer points out the important issue. Providing the detailed mechanism of MALDI-TOF-MS and LDI-TOF-MS analysis will be beneficial to the broad readership of Nanomaterials. Therefore, we have added sentences “The detailed mechanism of MALDI process is still not fully understood, but it has been generally described by serial 3-step processes including laser energy transfer from matrix to analyte in their solid state mixture upon laser irradiation, ionization by photochemical reaction and isolation of ionized analyte in excess matrix for mass spectrometric analysis [4].” and “Those sp2 carbon structures of GO derivatives play an important role in LDI-TOF-MS analysis by absorbing of laser energy and converting it into thermal energy through the electron-phonon interaction for LDI of small molecules [35].” to explain the mechanism of MALDI-TOF-MS and GO-based LDI-TOF-MS analysis in the revised version.
“Comment 2. Please describe the figures of merit regarding the performance of these matrices/platforms within the framework of mass spectroscopy.”
The reviewer raises a significant point. The figure of merit (FOM) of GO derivatives and their nanohybrid structures can provide meaningful information about their quantitative comparison as a platform for LDI-TOF-MS analysis of small molecules. We have described FOM, limit of detection (LOD), of the available analytical platforms composed of GO derivatives and their nanohybrid structures in the revised version. It is noteworthy that several references do not provide FOM but simple parallel comparison. The FOM values are also summarized in table 1.
“Comment 3. Please include a summary table with all the results and the figures of merits of the reviewed papers.”
We appreciate the reviewer’s comment. In the same line with answer to comment 2, the available FOM values from the reviewed papers are summarized in table 1 of the revised version with a proper description “As a summary of the important LDI-TOF-MS analytical platforms fabricated by using GO derivatives and their nanohybrid structures, the figure of merit (FOM) of those analytical platforms such as LOD values was described in Table 1. The summarized results indicated that the hybridization of CNT, metal and metal oxide nanomaterials on the surface of GO derivatives does not always guarantee the improvement of LOD values, but it can still endow their nanohybrid structures with a novel function and thus extend their analytical applicability according to the purposes.”. For the summary of FOM (Table 1), please see the attached file.
“Comment 4. I propose a SWOT analysis of the graphene-based matrices in mass spectroscopy as a potential technology.”
We thank the reviewer’s comment and have carried out a SWOT analysis on the basis of literature reviewed by this manuscript. The result of SWOT analysis is added as table 2 with a proper description “The strengths, weaknesses, opportunities and threats (SWOT) analysis was also carried out on the basis of the reviewed literature (Table 2). GO derivatives and their nanohybrid structures have exhibited many advantages including high laser energy absorption capacity, photothermal conversion efficiency, electrical and thermal conductivity, affinity to important biomolecules and environmental pollutants, and amenable surface for functionalization and hybridization with other functional groups and nanomaterials. However, they also possess disadvantages such as laser induced fragmentation, contamination of mass spectrometer, and heterogeneous lateral dimension and chemical structures.” to the revised version. For the result of SWOT analysis (Table 2), please see the attached file.
“Comment 5. The authors should include in the Conclusion paragraph their opinion for the perspectives and challenges of the technology they reviewed.”
We appreciate the reviewer’s comment and have added sentences “GO derivatives and their nanohybrid structures can provide distinct advantages and thus they will considerably contribute to various research fields including metabolomics, environmental pollution, imaging mass spectrometry and drug discovery. However, the potential toxicity, dangerous and toxic synthetic process and unstandardized structures of GO derivatives should be addressed for their wide-spread applications.” to the conclusion paragraph of the revised version to provide the perspectives and challenges of GO derivatives and their nanohybrid structures for LDI-TOF-MS analysis of small molecules
We are grateful to the reviewers and editor for their efforts. We believe that this work can provide a fundamental insight and practical information for studying and developing an efficient LDI-MS platform by using GO derivatives and their nanohybrid structures.
With very best regards,
Young-Kwan Kim

Round 2
Reviewer 1 Report
Most of the questions and suggestions are well addressed by author, however revision is needed
RAMAN spectroscopy of GO and rGO and its derivatives is a key characterization technique and it should not just based on D and G bands. References suggested in the revision by author are key bibliography for graphene materials, however, for highly defective graphene materials in stage 2 of defects and also in the transition between stage 2 and stage 1 of defects, there are more specific literature.
Broad D and G bands are associated to the presence of defects signals and are the resulting of the overlap of different interbands D’’, D, D* and G and D’ and a decrease in the defects concentration in the graphene materials are associated with a decrease in the FWHM of D and G bands in rGO. [see for example The Journal of Physical Chemistry C. 2015;119(18):10123-9. Materials Research Express. 2017;4(10):105020. Nano Letters. 2011;11(8):3190-6.]
ID/IG and IDD´/I2D bands relations are also key for the characterization of evolution from GO to rGO and the transition from stage 2 to stage 1 of defects. I I recommend again considering RAMAN and discussing it in this contex
Author Response
We thank to reviewer’s comment. We also think that Raman spectroscopy is a powerful and essential tool to characterize the defect structures in graphene derivatives and agree with the reviewer because Raman characterization is really important to reveal the structural nature of graphene derivatives. However, this review is mainly focused on the applications of GO derivatives and their nanohybrid structures to laser desorption/ionization time-of-flight mass spectrometry (LDI-TOF-MS) analysis of small molecules. Therefore, if we deeply deal with the Raman characterization of all cited references, it can distract the focus of this review and we want to emphasize that most of the cited references do not provide Raman spectra of GO derivatives and their nanohybrid structures which were utilized in their reports. Since Raman characterizations of GO derivatives and their nanohybrid structures are complicated as the reviewer pointed out “Broad D and G bands are associated to the presence of defects signals and are the resulting of the overlap of different interbands D’’, D, D* and G and D’ and a decrease in the defects concentration in the graphene materials are associated with a decrease in the FWHM of D and G bands in rGO.”, several references do not present and discuss the Raman spectral features in their main text. Furthermore, we have reviewed various GO derivatives and their nanohybrid structures. As the reviewer already knows, the GO derivatives are highly defective materials and their degree and kinds of defected structures are dependent on their synthetic process and starting graphite, and the nanohybridization also can affect their defect structures. Therefore, I think that Raman spectroscopic features of GO derivatives and their nanohybrid structures are too significant and complicated issue to be briefly included in this manuscript, and it is eligible to be reviewed as another review paper. Then again, I want to emphasize that the deep and comprehensive discussion of Raman spectroscopic analysis of GO derivatives and their nanohybrid structures is beyond the scope of this review and can also make the readers confused because of the limited information from the references. But, we also agree with the reviewer’s comment that Raman spectroscopy is particularly important to characterize the defect structures of GO derivatives and their nanohybrid structures.
To indicate this point, we have added a sentence “In this regard, Raman spectroscopy is a powerful and essential analytical tool to characterize the ordered and defected sp2 carbon structures of GO derivatives and their nanohybrid structures which are closely related to their electron-phonon transition and then their efficiency in LDI-TOF-MS analysis [42-44].” with the suggested references in the revised version.
We are grateful to the reviewers and editor for their efforts. We believe that this work can provide a fundamental insight and practical information for studying and developing an efficient LDI-MS platform by using GO derivatives and their nanohybrid structures.
With very best regards,
Young-Kwan Kim

Reviewer 3 Report
Accepted
Author Response
We appreciate your efforts and comments.